# THE INFINITE CONTEXTUAL GRAPH MARKOV MODEL

## ABSTRACT

The Contextual Graph Markov Model is a deep, unsupervised, and probabilistic model for graphs that is trained incrementally on a layer-by-layer basis. As with most Deep Graph Networks, an inherent limitation is the lack of an automatic mechanism to choose the size of each layer's latent representation. In this paper, we circumvent the problem by extending the Contextual Graph Markov Model with Hierarchical Dirichlet Processes. The resulting model for graphs can automatically adjust the complexity of each layer without the need to perform an extensive model selection. To improve the scalability of the method, we introduce a novel approximated inference procedure that better deals with larger graph topologies. The quality of the learned unsupervised representations is then evaluated across a set of eight graph classification tasks, showing competitive performances against end-to-end supervised methods. The analysis is complemented by studies on the importance of depth, hyper-parameters, and compression of the graph embeddings. We believe this to be an important step towards the theoretically grounded and automatic construction of deep probabilistic architectures for graphs.

## 1 INTRODUCTION

It can be argued that one of the most daunting processes in machine learning is the selection of appropriate hyper-parameters for the task at hand. Indeed, due to the data-dependent nature of the learning problem, there usually exists no single model configuration that works well in all contexts. The most straightforward approach to mitigate this issue has typically been to rely on standard model selection techniques such as grid and random searches (Bergstra & Bengio, 2012), where the range of values to try are fixed a priori by the user. Nonetheless, there has always been an interest in alternative methods that automatically choliteratureose the "right" values for some hyper-parameters (Gershman & Blei, 2012; He et al., 2021). In the Bayesian nonparametric (BNP) literature, which is of particular interest for this work, the complexity of Bayesian models automatically grows *with the data* (Teh et al., 2006), e.g., a BNP mixture model can adjust the number of its mixtures to better fit the empirical data distribution, thus freeing the user from the burden of choosing the most important (if not all) hyper-parameters.

In recent years, much research effort has been devoted to the theoretical and practical development of Deep Graph Networks (DGNs), which originated from Micheli (2009); Scarselli et al. (2009). DGNs can deal with graphs of varying topology without the need for human intervention, and they rely on local and iterative processing of information commonly known as *message passing*; for a thorough description of some of the most popular DGNs in the literature (and of the more general graph representation learning field) we refer the reader to recent surveys on the topic (Bronstein et al., 2017; Battaglia et al., 2018; Bacciu et al., 2020b; Wu et al., 2020). Despite most of these methods belonging to the neural world, the Contextual Graph Markov Model (CGMM) stands out as a deep, unsupervised, constructive and fully probabilistic model that has shown competitive performances on downstream graph classification tasks (Bacciu et al., 2018; 2020a). CGMM trains a stack of Bayesian networks, where each layer is conditioned on the *frozen* posteriors of the nodes of the graph computed at previous layers. Each layer optimizes the likelihood of the data using the Expectation Maximization (EM) algorithm (Moon, 1996) with closed-form solutions. Like its neural counterparts, for which the number of hidden units in each layer has typically been selected as a hyper-parameter, CGMM relies on model selection to choose the "reasonable" number of hidden states associated with the categorical latent variables. Differently from the neural methods though, CGMM is amenable to a BNP extension, as each layer is essentially a conditional mixture model.

The main challenge we tackle in this work is the adaptation of CGMM to the elegant theoretical framework of BNP methods, in order to automatize the choice of its hyper-parameters, e.g., the number of states. The principal difficulty lies in how to handle the variable-size number of neighbors of each node inside this framework, which in CGMM is solved by (possibly weighted) convex combinations of the neighbors' posteriors. The resulting model, called Infinite Contextual Graph Markov Model (ICGMM), can generate as many latent states as needed to solve the unsupervised density estimation task at each layer. To the extent of our knowledge, this is the first Bayesian nonparametric model for adaptive graph processing. As a second contribution, we provide a faster implementation of our method that scales to the social datasets considered in this work while still providing state of the art results. We compare ICGMM against CGMM as well as end-to-end supervised methods on eight different graph classification tasks, following a fair, robust and reproducible experimental procedure (Errica et al., 2020). Results show that ICGMM performs on par or better than the related models. We complement the analysis with studies on the effects of depth and generation of our model's latent states. All in all, we believe that ICGMM is an important (if not the first) step towards a theoretically grounded and automatic construction of Deep Bayesian Graph Networks.

## 2 RELATED WORKS

The fundamental Bayesian nonparametric literature that is relevant to our work relates to the families of Dirichlet Processes (DPs) (Gershman & Blei, 2012) and Hierarchical Dirichlet Processes (HDPs) (Teh et al., 2006). In its most essential definition, a DP is a stochastic process that defines a probability distribution over other probability distributions. A DP is parametrized by a base distribution $G_0$, i.e., the expected value of the process, and a scaling parameter $\alpha_0$ that controls the concentration of DP realizations around $G_0$ (Teh, 2010). In particular, the Chinese Restaurant Process (Aldous, 1985), the Stick-breaking Construction (Sethuraman, 1994) and the Pòlya urn scheme (Hoppe, 1984) are all alternative ways to formalize a DP. Moving to HDPs is conceptually straightforward, in that it considers the base distribution $G_0$ as a draw from another DP parametrized by a base distribution $H$ and a scaling parameter $\gamma$. For a detailed treatment of learning with DP and HDPs, the reader can check a number of tutorials and surveys (Teh et al., 2006; Orbanz & Teh, 2010; Gershman & Blei, 2012). Our work shares similarities with the Infinite Hidden Markov Model for temporal series (Beal et al., 2002), with the fundamental differences that causality assumptions have to be relaxed to deal with graphs and that the hidden variables' distributions are conditioned on a varying number of observations.

Most of the recent advances of the graph representation learning field are based on the so called feedforward DGNs (Bacciu et al., 2020b). These models rely on "spatial" graph convolutional layers, i.e., the state of each node in the graph is determined by applying a permutation invariant function to its neighboring states computed at the previous layers. Combined with the depth of the architecture, these models propagate contextual information across the graph, a process also known as "message passing" (Gilmer et al., 2017). However, to the best of our knowledge, the only neural method for graphs that automatically constructs part of its architecture in a principled way is the pioonering work of Micheli (2009). In fact, the Neural Network for Graphs (NN4G), known to be the first spatial DGN, relies on the Cascade Correlation learning algorithm (Fahlman & Lebiere, 1990) to determine the number of layers to use for the task under investigation.

Instead, despite being loosely related to our work, AutoML methods for graphs are yet another way to automatize the selection of all hyper-parameters of a DGN (He et al., 2021). In particular, the Auto-GNN technique relies on Neural Architecture Search to discover, based on performance trends, an adequate configuration for the supervised task (Zhou et al., 2019). We differ from these approaches in two fundamental respects: first, we build upon theoretical grounds rooted in the BNP literature; secondly, we determine the right number of states in a completely unsupervised fashion.

In what follows, we provide a formalization of the Infinite Contextual Graph Markov Model. Apart from the technical details, our hope is to show how the cross-fertilization of ideas from different research fields can help us advance the state of the art, both in the theoretical and empirical sense.

## 3 METHOD

This Section introduces the details of our method. Since we borrow ideas from two relatively distant fields, we define a unified mathematical notation and jargon as well as a high-level overview of the CGMM and HDP models to ease the subsequent exposition.

We define a graph as a tuple $g = (\mathcal{V}_g, \mathcal{E}_g, \mathcal{X}_g)$ where $\mathcal{V}_g$ is the set of entities (also referred to as nodes or vertices), $\mathcal{E}_g$ is the set of oriented edges $(u, v)$ connecting node $u$ to $v$, and the symbol $\mathcal{X}_g$ stands for the set of node attributes associated with the graph $g$. Also, the neighborhood of a node $u$ is the set of nodes connected to $u$, i.e., $\mathcal{N}_u = \{v \in \mathcal{V}_g | (v, u) \in \mathcal{E}_g\}$. For the purpose of this work, we will define the (categorical or continuous) node feature of a node $u$ with the term $x_u \in \mathcal{X}_g$.

### 3.1 BASICS OF CGMM

To best understand how and why this work extends CGMM, we now give a brief but essential description of its main characteristics. CGMM is, first and foremost, a deep architecture for the adaptive processing of graphs. Like other DGNs, it maps the entities of a graph, if not the graph itself, into latent representations. More specifically, we can get one of such representations for each layer of the architecture and then concatenate all of them to obtain richer node and graph embeddings. The latter is usually obtained as a global aggregation of the former.

The second peculiarity of CGMM is that it is constructive, i.e., trained in an incremental fashion: after one layer is trained, another one can be stacked atop of it and trained using the frozen outputs of the previous layer. This idea is borrowed from NN4G (Micheli, 2009), and it allows CGMM to relax the mutual dependencies between latent variables in a cyclic graph. However, because the local and iterative message passing mechanism used by spatial methods (Micheli, 2009; Kipf & Welling, 2017) is responsible for information propagation across the graph, this relaxation is not restrictive.

Thirdly, the node/graph embedding construction is fully probabilistic and unsupervised, since layer $\ell$ is represented as the Bayesian network on the left hand-side of Figure 1. A latent variable $q_u^\ell$ is attached to each node $u$, and it is responsible for the the generation of the node feature $x_u$. To take into account structural information, the hidden state $q_u^\ell$ is conditioned on the neighboring hidden states computed at the previous layer, i.e., the set $\{q_v^{\ell-1} \mid v \in \mathcal{N}_u\}$. Importantly, the constructive approach allows us to treat the hidden (frozen) states of the previous layer as observable variables. Each layer is trained to fit the data distribution of node features using the EM algorithm, thus guaranteeing the convergence to a local minima. Once inference is performed, the state of each node is frozen and we can move to the subsequent layer. Lastly, the embedding of each node at layer $\ell$ is encoded as the posterior of its hidden state.

### 3.2 BASICS OF HDP

The HDP is a Bayesian nonparametric prior for the generation of grouped data using different infinite mixture models with shared mixture components. Let $\{x_1, x_2, \dots\}$ be a set of observations that are grouped into $J$ groups, i.e., each observation $x_u$ belongs to the group $j_u \in \{1, \dots, J\}$. Using the stick-breaking representation (Sethuraman, 1994), the HDP mixture model that generates the observations can be defined as (Teh et al., 2006):

$$
\begin{aligned}
\boldsymbol{\beta} \mid \gamma &\sim \mathrm{Stick}(\gamma) & q_u \mid j_u, (\boldsymbol{\pi}_j)_{j=1}^J &\sim \boldsymbol{\pi}_{j_u} \\
\boldsymbol{\pi}_j \mid \boldsymbol{\beta}, \alpha_0 &\sim \mathrm{DP}(\alpha_0, \boldsymbol{\beta}) & x_u \mid q_u, (\boldsymbol{\theta}_c)_{c=1}^\infty &\sim F(\boldsymbol{\theta}_{q_u}) \\
\boldsymbol{\theta} \mid \boldsymbol{H} &\sim \boldsymbol{H},
\end{aligned}
\tag{1}
$$

where $F(\boldsymbol{\theta}_{q_u})$ denotes the emission distribution, parametrized by $\boldsymbol{\theta}_{q_u}$, that generates the observation $x_u$. The latent state $q_u$ indicates which mixture component should be used to generate $x_u$. The value of $q_u$ is sampled from the distribution $\boldsymbol{\pi}_{j_u}$, which stands for the mixture weights of group $j_u$. All $(\boldsymbol{\pi}_j)_{j=1}^J$ are obtained from a DP with concentration parameter $\alpha_0$ and base distribution $\beta$. Notably, all groups' mixture weights are defined on the same set of mixture components, meaning there is a form of parameter sharing across different groups. Finally, we sample the distribution $\boldsymbol{\beta}$ via the stick-breaking process $\mathrm{Stick}(\gamma)$ of Sethuraman (1994).

To generate a possibly infinite number of emission distributions, we exploit a prior distribution $\boldsymbol{H}$ that allows us to create new mixture components on demand. Thanks to the stick-breaking construction,

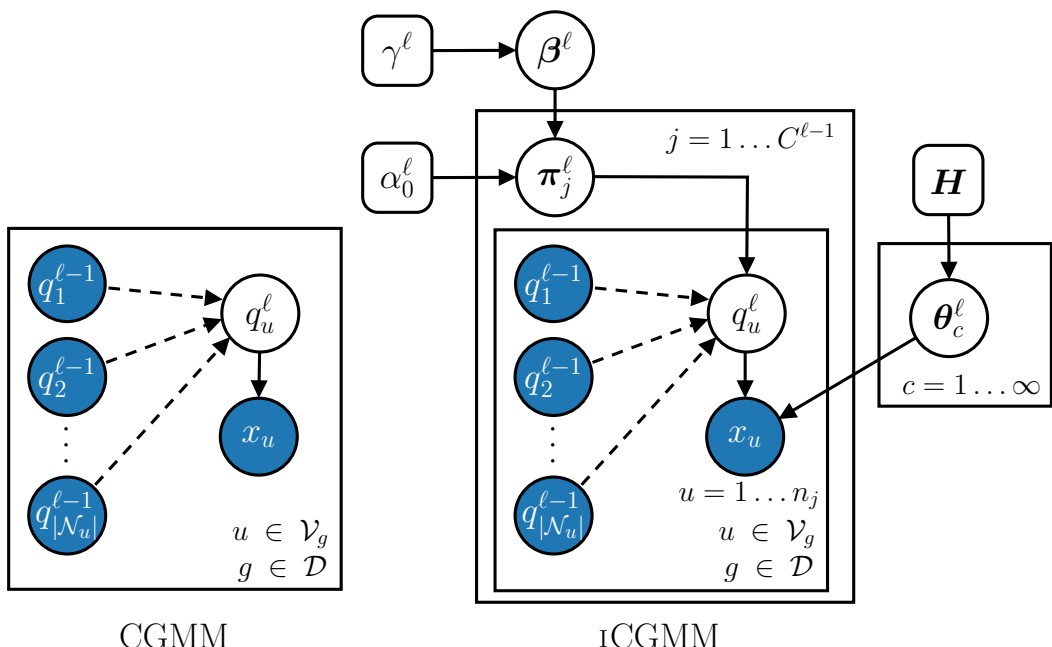

CGMM                    ɪCGMM

Figure 1: Differences between layer $\ell$'s graphical model of the original CGMM and the proposed ɪCGMM. Observable variables are blue circles, latent ones are empty circles, and white boxes denote prior knowledge. Each ɪCGMM is an HDP mixture model where the group $j$ for each node observation $x_u$ is pre-determined by the set of states of neighboring nodes $\mathbf{q}_{\mathcal{N}_u}^{\ell-1}$ computed at layer $\ell-1$. Contrarily to CGMM, the number of values that the latent indicator variable $q_u$ can assume is automatically adjusted to fit the underlying data distribution. Dashed arrows denote the flow of contextual information from previous layers through the neighbors of each node $u$.

even though an infinite number of mixture components can be used, only a finite number of them is istantiated during the inference phase. Hereinafter, we indicate with the symbol $C$ the number of mixture components that are chosen by the HDP at inference time.

### 3.3 Model Definition

Architecturally speaking, ɪCGMM shares the same characteristics of CGMM described in Section 3.1, whereas the differences of each layer's graphical model are highlighted in Figure 1. In particular, ɪCGMM assumes that the generation of the node features $x_u$ at each layer is governed by a HDP mixture model. Thus, following the stick-breaking construction detailed in Section 3.2, the generative process of a single ɪCGMM layer $\ell$ can be formalized as follows:

$$
\begin{aligned}
\boldsymbol{\beta}^\ell \mid \gamma^\ell &\sim \text{Stick}(\gamma^\ell) & j_u^\ell \mid \mathbf{q}_{\mathcal{N}_u}^{\ell-1} &= \psi(\mathbf{q}_{\mathcal{N}_u}^{\ell-1}) \\
\boldsymbol{\pi}_j^\ell \mid \boldsymbol{\beta}^\ell, \alpha_0^\ell &\sim \text{DP}(\alpha_0^\ell, \boldsymbol{\beta}^\ell) & q_u^\ell \mid j_u^\ell, (\boldsymbol{\pi}_j)_{j=1}^{C^{\ell-1}} &\sim \boldsymbol{\pi}_{j_u}^\ell & (2) \\
\boldsymbol{\theta}^\ell \mid \boldsymbol{H} &\sim \boldsymbol{H} & x_u \mid q_u^\ell, (\boldsymbol{\theta}_c^\ell)_{c=1}^\infty &\sim F(\boldsymbol{\theta}_{q_u^\ell}^\ell),
\end{aligned}
$$

where we add the superscript $\ell$ to the HDP mixture model quantities to highlight that they are different at each ɪCGMM layer. Similarly to the HDP case, we use $C^\ell$ to denote the number of states chosen by the model at the current layer. When clear from the context, we will omit such a superscript to ease the notation.

In any HDP mixture model, each observation must be assigned to a group *in advance*. In this work, the assignment of a node feature to a group is not known a priori, but that is the key to propagate contextual information across the graph. We select the group $j_u^\ell$ of the feature node $x_u$ based on the neighbors' observable posteriors computed at the previous layer, i.e., $\mathbf{q}_{\mathcal{N}_u}^{\ell-1} = \{\boldsymbol{q}_v^{\ell-1} \in [0,1]^{C^{l-1}} \mid v \in \mathcal{N}_u\}$. However, due to known intractability issues, each posterior distribution $\boldsymbol{q}_v^{\ell-1}$ is approximated by the

inference phase of the previous layer (see Section 3.4). To stick as much as possible to the original CGMM formalism, $j_u$ is chosen as the most likely position in the $C^{\ell-1}$-sized macrostate, obtained by averaging the neighbors' probabilities in $\mathbf{q}_{\mathcal{N}_u}^{\ell-1}$:

$$j_u^\ell = \psi(\mathbf{q}_{\mathcal{N}_u}^{\ell-1}) = \underset{j \in \{1,\dots,C^{\ell-1}\}}{\arg\max} \Big( \frac{1}{|\mathcal{N}_u|} \sum_{v \in \mathcal{N}_u} \boldsymbol{q}_v^{\ell-1} \Big)_j. \tag{3}$$

It follows that nodes with the same feature may have a different latent state $c$, due to the fact that they are assigned to different groups, i.e., different $\boldsymbol{\pi}_j$, on the basis of their neighborhood; this mimics the role of CGMM neighborhood aggregation but in an HDP mixture model. In the first layer, where no previous layer exists, we shall just assume that all node features belong to the same group.

Summing up, we depart from the basic CGMM layer of Bacciu et al. (2020a) in more than one way. First and foremost, we do not parametrize nor learn the CGMM transition distribution, which was responsible for the convex combination of neighboring states when computing the E-step of the EM algorithm. Instead, we rely on the most probable choice of the group $j_u$ that is encoded by the neighbors' macrostate. Secondly, due to the sheer complexity of the Bayesian nonparametric treatment, we do not train the model via EM as done with CGMM; instead, we will exploit Gibbs sampling (Geman & Geman, 1984) to compute the quantities of interest. Finally, ICGMM retains one important architectural characteristic of CGMM, i.e., it prevents vanishing gradient effects and over-smoothing by default (Bacciu et al., 2020a), thus allowing us to construct deeper architectures that propagate contextual information.

### 3.4 INFERENCE

The inference phase of every BNP method is meant to estimate the posterior of the model parameters. For each ICGMM layer $\ell$, we wish to compute the quantities $\boldsymbol{q}_u^\ell, \boldsymbol{\beta}^\ell, \boldsymbol{\pi}_j^\ell$ and $\boldsymbol{\theta}^\ell$. Thanks to the incremental construction of the ICGMM architecture, we can do so one layer at a time. It is worth mentioning that the constructive approach of CGMM is not an approximation of a more complex graphical model, rather it is a design choice that applies the basic principle of iterative computation underpinning all DGNs. Thus, since each ICGMM layer is an HDP mixture model, we can infer its parameters following the Gibbs sampling schema of Teh et al. (2006). Note that it is also possible to estimate the hyper-parameters $\alpha_0^\ell$ and $\gamma^\ell$: whenever that is the case, we shall append a subscript *"auto"* to our model's name. In the interest of space, we report the ICGMM complete Gibbs sampling equations and pseudo-code in Appendix A and B, respectively.

**Graph Embedding Generation.**   In a similar vein with (Bacciu et al., 2020a), we prefer to use the sample distribution of $q_u$ (Eq. 4) at the last iteration, rather than the last sampled state, as an approximation of node $u$'s posterior distribution. This way, we encode more information about state occupancy into node/graph embeddings.

As in Bacciu et al. (2020a), node embeddings of each layer are represented as *unibigrams*. A unibigram concatenates the posterior of a node, i.e., a vector called *unigram*, with its *bigram*. A bigram counts, for each possible state $q_u$, how many of $u$'s neighbors are in another state, and it is represented as a vector of size $C^2$. The final graph representation that is fed into the classifier is obtained by concatenation of node unibigrams across all layers followed by global aggregation.

**Faster Inference with Node Batches (ICGMM$_f$).**   Due to the sequential nature of the Gibbs sampling procedure, a naive implementation is slow when applied to the larger social graphs considered in this work. In the literature, there exist several exact distributed inference methods for HDP (Lovell et al., 2012; Williamson et al., 2013; Chang & Fisher III, 2014; Ge et al., 2015), but their effectiveness might be limited due to the unbalanced workload among workers or the elevated rejection rate (Gal & Ghahramani, 2014).

In this work, we prefer to speed-up the inference procedure by introducing an approximation rather than relying on an exact distributed computation. As suggested in Gal & Ghahramani (2014), an approximated inference procedure may indeed suffice for many problems. What we propose is based on a straightforward idea, which is to perform sampling for a batch of node observations in parallel. This way, the necessary statistics are updated in batch rather than individually, and matrix operations can be used to gain efficiency. To maintain a good trade-off between the quality and speedup, we stick

to 1 graph as the size of our batch. Such a trade-off provides a CPU speedup of up to $60\times$ at training time, and we empirically observed that performances remain unchanged with respect to the original version on the smaller chemical tasks considered. While this faster version of ɪCGMM, which we call ɪCGMM$_f$, does not strictly adhere to the technical specifications of the previous section, we believe that the pros largely outperform the cons. The interested reader can refer to Appendix D for an analysis of the speedup gains on the different datasets.

### 3.5 LIMITATIONS

Due to the complexity of the BNP treatment, one limitation of this work is that naive Gibbs sampling does not scale to very large datasets. The node independence assumption made by CGMM enables a faster batch computation, which can also be run on GPU. Despite having provided a simple, but approximated, sampling process that guarantees a substantial speedup and allows us to process graphs of non-negligible size, it would be interesting in the future to explore other inference methods to increase ɪCGMM's speedup, e.g., variational inference (Bryant & Sudderth, 2012; Wang & Blei, 2012; Hoffman et al., 2013; Hughes et al., 2015). The second limitation of ɪCGMM is that edge features are not taken into account. While there exist many neural models that do the same, we know that CGMM and its variant E-CGMM (Atzeni et al., 2021) can deal with discrete and arbitrary features, respectively. Our research directions for the future will investigate these aspects, providing an exact and efficient version of ɪCGMM that can process edge features as well.

## 4 EXPERIMENTS

We evaluated the performances of ɪCGMM using the fair, robust, and reproducible evaluation setup for graph classification defined in Errica et al. (2020). It consists of an external 10-fold cross validation for model assessment, followed by an internal hold-out model selection for each of the external folds. Stratified data splits were already provided; in this respect, we had to re-assess CGMM and E-CGMM (Atzeni et al., 2021), a recently proposed variant, by trying all the hyper-parameters specified in the original papers (in particular, the values of $C$ tried were 5, 10 and 20). We first experiment on the three chemical datasets D&D (Dobson & Doig, 2003), NCI1 (Wale et al., 2008) and PROTEINS (Borgwardt et al., 2005), where node features represent atom types. Then, we consider social datasets, including IMDB-BINARY, IMDB-MULTI, REDDIT-BINARY, REDDIT-MULTI-5K, and COLLAB (Yanardag & Vishwanathan, 2015), where the degree of each node is the sole continuous feature available. All datasets are publicly available (Kersting et al., 2016) and their statistics are summarized in Appendix C. Finally, we relied on Pytorch Geometric (Fey & Lenssen, 2019) for the implementation of our method.[1]

Apart from CGMM's variants, we will compare ɪCGMM against the following end-to-end supervised neural architectures for graphs: DGCNN (Zhang et al., 2018), DIFFPOOL (Ying et al., 2018), ECC (Simonovsky & Komodakis, 2017), GIN (Xu et al., 2019), GRAPHSAGE (Hamilton et al., 2017), and a structure-agnostic baseline method BASELINE, described in Errica et al. (2020), which was competitive on a number of benchmarks. We recall that these supervised methods construct the graph embeddings leveraging the supervision information coming from the target label; on the contrary, ɪCGMM embeddings are built in an unsupervised and constructive way, thus this represents a challenging comparison for our approach. Results for the supervised models are taken from (Errica et al., 2020).

We have discussed how ɪCGMM can automatize the choice its hyper-parameters, e.g., the size of the latent representation. In general, the choice of the Bayesian hyper-parameters is much less important than that of the number of states $C$, as in principle one can recursively introduce hyper-priors over these hyper-parameters (Bernardo & Smith, 2009; Goel & Degroot, 1981). That said, being this the first work to study HDP methods in the context of graph classification, we both i) explored the hyper-parameter space to best assess and characterize the behaviour of the model and ii) introduced hyper-priors to estimate $\alpha_0^\ell$ and $\gamma^\ell$ at each layer, thus further reducing the need for an extensive model selection.

For the chemical tasks, the prior $\boldsymbol{H}$ over the emission parameters $\boldsymbol{\theta}_c$ was the uniform Dirichlet distribution. The range of ɪCGMM hyper-parameters tried in this case were: number of layers

---

[1]The code to rigorously reproduce our results is provided in the supplementary material.

Table 1: Results on chemical datasets (mean accuracy and standard deviation) are shown. The best performances are highlighted in bold.

|  | D&D | NCI1 | PROTEINS |
|---|---|---|---|
| BASELINE | $\mathbf{78.4} \pm 4.5$ | $69.8 \pm 2.2$ | $\mathbf{75.8} \pm 3.7$ |
| DGCNN | $76.6 \pm 4.3$ | $76.4 \pm 1.7$ | $72.9 \pm 3.5$ |
| DIFFPOOL | $75.0 \pm 3.5$ | $76.9 \pm 1.9$ | $73.7 \pm 3.5$ |
| ECC | $72.6 \pm 4.1$ | $76.2 \pm 1.4$ | $72.3 \pm 3.4$ |
| GIN | $75.3 \pm 2.9$ | $\mathbf{80.0} \pm 1.4$ | $73.3 \pm 4.0$ |
| GRAPHSAGE | $72.9 \pm 2.0$ | $76.0 \pm 1.8$ | $73.0 \pm 4.5$ |
| CGMM | $74.9 \pm 3.4$ | $76.2 \pm 2.0$ | $74.0 \pm 3.9$ |
| E-CGMM | $73.9 \pm 4.1$ | $78.5 \pm 1.7$ | $73.3 \pm 4.1$ |
| ICGMM | $75.6 \pm 4.3$ | $76.5 \pm 1.8$ | $72.7 \pm 3.4$ |
| ICGMM$_f$ | $75.0 \pm 5.6$ | $76.7 \pm 1.7$ | $73.3 \pm 2.9$ |
| ICGMM$_{auto}$ | $76.3 \pm 5.6$ | $77.6 \pm 1.5$ | $73.1 \pm 3.9$ |
| ICGMM$_{f_{auto}}$ | $75.1 \pm 3.8$ | $76.4 \pm 1.4$ | $73.2 \pm 3.9$ |

Table 2: Results on social datasets (mean accuracy and standard deviation) are shown, where the node degree is used as the only node feature. The best performances are highlighted in bold.

|  | IMDB-B | IMDB-M | REDDIT-B | REDDIT-5K | COLLAB |
|---|---|---|---|---|---|
| BASELINE | $70.8 \pm 5.0$ | $\mathbf{49.1} \pm 3.5$ | $82.2 \pm 3.0$ | $52.2 \pm 1.5$ | $70.2 \pm 1.5$ |
| DGCNN | $69.2 \pm 3.0$ | $45.6 \pm 3.4$ | $87.8 \pm 2.5$ | $49.2 \pm 1.2$ | $71.2 \pm 1.9$ |
| DIFFPOOL | $68.4 \pm 3.3$ | $45.6 \pm 3.4$ | $89.1 \pm 1.6$ | $53.8 \pm 1.4$ | $68.9 \pm 2.0$ |
| ECC | $67.7 \pm 2.8$ | $43.5 \pm 3.1$ | - | - | - |
| GIN | $71.2 \pm 3.9$ | $48.5 \pm 3.3$ | $89.9 \pm 1.9$ | $\mathbf{56.1} \pm 1.7$ | $75.6 \pm 2.3$ |
| GRAPHSAGE | $68.8 \pm 4.5$ | $47.6 \pm 3.5$ | $84.3 \pm 1.9$ | $50.0 \pm 1.3$ | $73.9 \pm 1.7$ |
| CGMM | $72.7 \pm 3.6$ | $47.5 \pm 3.9$ | $88.1 \pm 1.9$ | $52.4 \pm 2.2$ | $77.32 \pm 2.2$ |
| E-CGMM | $70.7 \pm 3.8$ | $48.3 \pm 4.1$ | $89.5 \pm 1.3$ | $53.7 \pm 1.0$ | $77.45 \pm 2.3$ |
| ICGMM$_f$ | $\mathbf{73.0} \pm 4.3$ | $48.6 \pm 3.4$ | $91.3 \pm 1.8$ | $55.5 \pm 1.9$ | $78.6 \pm 2.8$ |
| ICGMM$_{f_{auto}}$ | $71.8 \pm 4.4$ | $49.0 \pm 3.8$ | $\mathbf{91.6} \pm 2.1$ | $55.6 \pm 1.7$ | $\mathbf{78.9} \pm 1.7$ |

$\in \{5, 10, 15, 20\}$, $\alpha_0 \in \{1, 5\}$, $\gamma \in \{1, 2, 3\}$, unigram aggregation $\in \{\mathrm{sum}, \mathrm{mean}\}$, and Gibbs sampling iterations $\in \{10, 20, 50\}$. Instead, for the social tasks we implemented a Normal-Gamma prior $\boldsymbol{H}$ over a Gaussian distribution. Here the prior is parametrized by the following hyper-priors: $\mu_0$, the mean node degree extracted from the data; $\lambda_0$, which is inversely proportional to the prior variance of the mean; and $(a_0, b_0)$, whose ratio $t = \frac{b_0}{a_0}$ represents the expected variance of the data. The ICGMM hyper-parameters here were: number of layers $\in \{5, 10, 15, 20\}$, $\lambda_0 \in \{\text{1e-6}\}$, $a_0 \in \{1.\}$, $b_0 \in \{0.09, 1.\}$, $\alpha_0 \in \{1, 5, 10\}$, $\gamma \in \{2, 5, 10\}$, unigram aggregation $\{\mathrm{sum}, \mathrm{mean}\}$, and Gibbs Sampling iterations $\in \{100\}$. To further automate learning of ICGMM's unsupervised layers, we place uninformative $Gamma(1, rate = 0.01)$ hyper-priors on both $\alpha_0^\ell, \gamma^\ell$ hyper-parameters. To prevent the model from getting stuck in a local minimum on COLLAB (due to bimodal degree distribution and large variances), we tried $\lambda_0 \in \{\text{1e-4}, \text{1e-5}\}$.

To conclude, we list the hyper-parameters tried for the one-layer MLP classifier trained on the unsupervised graph embeddings: optimizer $\in \{\mathrm{Adam}\}$, batch size $\in \{32\}$, hidden units $\in \{32, 128\}$, learning rate $\in \{\text{1e-3}\}$, L2 regularization $\in \{0., \text{5e-4}\}$, epochs $\in \{2000\}$, ReLU activation, and early stopping on validation accuracy with patience 300 on chemical tasks and 100 on social ones.

## 5 RESULTS

The empirical results on chemical and social benchmarks are reported in Tables 1 and 2, respectively. There are several observations to be made, starting with the chemical tasks. First of all, ICGMM performs similarly to CGMM, E-CGMM, and most of the *supervised* neural models; this suggests that the selection of $j_u$ based on the neighboring recommendations is a subtle but effective form of information propagation between the nodes of the graph. In addition, results indicate that we have succeeded in effectively automatizing the choice of the number of latent states without compromising the accuracy, which was the main goal of this work. Finally, ICGMM$_f$ performs as well as the exact version, and for this reason we safely applied the faster variant to the larger social datasets (including IMDB-B and IMDB-M to ease the exposition).

Moving to the social datasets, we observe that ɪCGMM achieves better average performances than other methods on IMDB-B, REDDIT-B and COLLAB. One possible reason for such an improvement with respect to CGMM variants may be how the emission distributions are initialized. On the one hand, and differently from the chemical tasks, CGMM and E-CGMM use the $k$-means algorithm (with fixed $k=C$), to initialize the mean values of the $C$ Gaussian distributions, which can be stuck in a local minimum around the most frequent degree values. One the other hand, ɪCGMM adopts a fully Bayesian treatment, which combined with the automatic selection of the latent states allows to better model outliers by adding a new state when the posterior probability of a data point is too low.

In what follows, we will try to shed more light into the improved generalization performances of ɪCGMM, by analyzing the exact model from a layer-wise perspective.

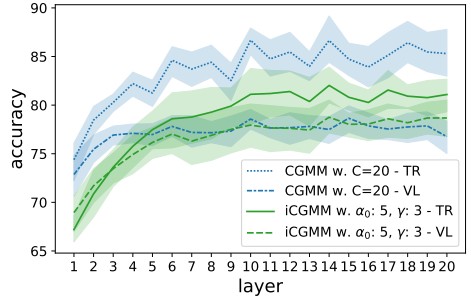

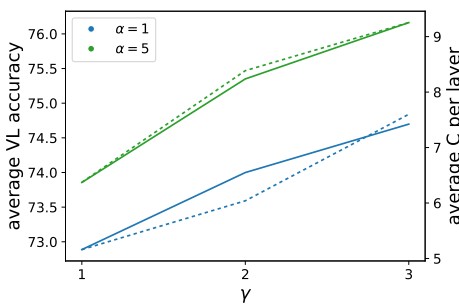

(a) Effect of depth on training/validation accuracy

(b) Average VL accuracy (solid line) and number of chosen states (dashed line) w.r.t $\alpha_0$ and $\gamma$ values

Figure 2: Figures 2a and 2b analyze the relation between depth, performances, and the number of chosen states on NCI1.

**On the effectiveness of depth and hyper-parameters.**    To confirm our intuition about the benefits of the proposed information propagation mechanism, Figure 2a shows the NCI1 training and validation performances of both CGMM and ɪCGMM as we add more layers. For simplicity, we picked the best ɪCGMM configuration on the first external fold, and we compared it against the CGMM configuration with the most similar performances. Note that $C = 20$ was the most frequent choice of CGMM states by the best model configurations across the 10 outer folds: this is because having more emission distributions to choose from allows the CGMM model to find better local minima, whereas ɪCGMM can automatically add states whenever the data point's sampling probabilities are too low. We trained the same classifier at different depths, and we averaged scores across the 10 outer folds. We observe that the validation performance of both models are similar, with an asymptotic behavior as we reach 20 layers; it follows that depth remains fundamental to improve the generalization performances (Bacciu et al., 2020a). Importantly, we see that gap between ɪCGMM training and validation scores is thinner than its non-BNP counterpart, suggesting that the classifier is less prone to overfitting the data.

We now study how ɪCGMM behaves as we vary the main hyper-parameters $\alpha_0$ and $\gamma$. We continue our experimentation on NCI1; Figure 2b depicts the average validation performance and number of states $C$ over all configurations and folds, subject to changes of $\alpha_0$ and $\gamma$ values. The trend indicates how greater values for both hyper-parameters achieve, on average, better validation performance. Also, smaller values of the two hyper-parameters tend to strongly regularize the model by creating fewer states, with consequent reduction in validation accuracy. The relation between the number of states and these hyper-parameters remains consistent with the mathematical details of Section 3.

**On the quality of graph embeddings.**    So far, we have argued that ɪCGMM selects the appropriate number of states for its unsupervised task at each layer. As a matter of fact, Figure 3a reports such a statistic on the same NCI1 configuration as before: ɪCGMM preferred a lower number of latent states than CGMM, i.e., around 5 per layer. In turn, the resulting graph embeddings become much smaller, with important savings in terms of memory footprint and computational costs to train the subsequent classifier. Figure 3b displays the cumulative graph embedding size across layers, using

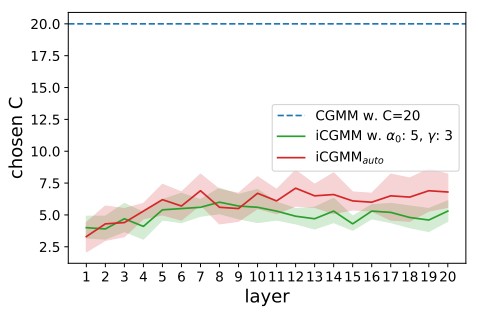

(a) # states chosen at each layer

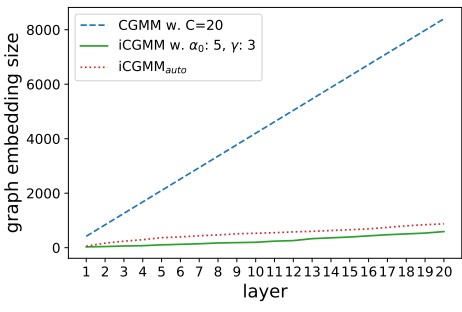

(b) Cumulative graph embedding size on NCI1

Figure 3: We show comparative results on the size and quality of graph embeddings between CGMM and ICGMM. Overall, ICGMM generates $\approx 0$ unused latent states, with consequent savings in terms of memory and compute time of the classifier with respect to CGMM. See the text for more details.

the unibigram representation without loss of generality. We see that, when compared with CGMM ($C$=20), the size of graph embeddings produced by ICGMM is approximately 7% of those of the original model, while still preserving the same performance as CGMM.

**On the automatic estimation of $\alpha^\ell$ and $\gamma^\ell$.** We conclude this work with a performance analysis of the fully automated versions of ICGMM and ICGMM$_f$, namely those with an *"auto"* subscript in Tables 1 and 2; in particular, we observe no statistically significant performance differences with respect to the original models. By estimating all hyper-parameters of our models using uninformative priors, we almost always (but for COLLAB) managed to *avoid the model selection* for the unsupervised graph embeddings creation. In turn, this amounted to a $6\times$ reduction in the overall number of configurations to be tried, but most importantly it frees the user from making hard choices about which configurations of hyper-parameters to try. Additionally, we observe that the number of chosen states and the consequent graph embedding size is very similar to that of ICGMM with $\alpha_0 = 5, \gamma = 3$, but this time the two hyper-parameters have been automatically adjusted by the model on the basis of the data.

To sum up, we have shown that: i) our model has very competitive performances with respect to the state of the art; ii) the information propagation mechanism introduced in the HDP is effective; iii) the model can automatically selects the number of states; iv) we can get a much lower memory and computational footprints due to the previous points without sacrificing the predictive performance; v) we can fully automatize the choice of the hyper-parameters using uninformative priors, which drastically reduces the cost of the model selection phase.

## 6 CONCLUSIONS

With the Infinite Contextual Graph Markov Model, we have bridged the gap between Bayesian nonparametric techniques and machine learning for graphs. We have described how our approach can automatically adjust the number of states and most hyper-parameters of each unsupervised layer, thus freeing the user from the burden of selecting them a priori. As the empirical analyses show, not only can the model exploit depth to increase its generalization performances, but it also produces smaller embeddings than CGMM, with consequent savings in terms of memory footprint and training time of the subsequent classifier. For these reasons, we believe that ICGMM represents a first relevant step towards the automatic construction of fully probabilistic deep learning models for graphs.

## REPRODUCIBILITY STATEMENT

To ensure that the results in this paper are reproducible, we relied on the PyDGN library (`https://pypi.org/project/PyDGN/`), which automatically handles both data and experiment pipelines, thus letting the researcher focus on the model definition. We follow the robust and reproducible settings of Errica et al. (2020); Section 4 reports further experimental details and the hyper-parameters tried for the models considered. Likewise, the appendix contains a detailed description of the inference procedure, as well as the pseudocode that has been implemented in the supplementary material. The data splits used can be retrieved from Errica et al. (2020), but they are nonetheless stored in the supplementary material alongside the code for ICGMM. To completely reproduce the experiments (model selection, model assessment) and the data pre-processing steps, we have also provided the necessary PyDGN configuration files.

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

# A ICGMM GIBBS SAMPLING PROCEDURE

HDP Gibbs sampling is an iterative procedure (Neal, 2000; Teh et al., 2006; Fox et al., 2007) that we use to estimate all node latent states and ICGMM's parameters at each layer. Hereinafter, to keep the notation less cluttered, we shall omit the superscript $\ell$ of the current layer and define $\bar{C}=C^{\ell-1}$.

**Sampling $q_u$.** The conditional distribution $\boldsymbol{q}_u$ of $q_u$ given all the other variables is given by:

$$P(q_u = c \mid j_u = j, \boldsymbol{q}^{-u}, \boldsymbol{\beta}, \boldsymbol{\theta}, \boldsymbol{x}) \propto (\alpha_0 \beta_c + n_{jc}^{-u}) f(x_u \mid \theta_c), \quad c \in \{1, \ldots, C+1\}; \quad (4)$$

where we recall that $C$ denotes the number of current states in the mixture model, $f$ is the *p.d.f.* associated with emission distribution $F(\theta)$ and the distribution $\boldsymbol{\pi}_j$ has been integrated out (Teh et al., 2006). Here, $n_{jc}^{-u}$ indicates the number of observations assigned so far to the mixture component $c$ of group $j$. Whenever we have that $q_u = C+1$, we create a new state and sample a new emission distribution $\boldsymbol{\theta}_{C+1}$ from $\boldsymbol{H}$. On the contrary, if at the end of an iteration there are no observation of any group associated with a certain mixture component, we can remove that mixture component and decrement the current number of states $C$. This is how ICGMM varies in complexity to fit the data distribution. Also, note that $\boldsymbol{q}_u^\ell$ will be used in Eq. 3 at the next layer $\ell+1$. When inferring the latent states of a new data point, no statistics of the model are updated.

**Sampling $\boldsymbol{\beta}$.** In the HDP stick-breaking representation that we use to define the ICGMM in Section 3.3, we require an auxiliary variable method to sample the base distribution $\boldsymbol{\beta}$ (Teh et al., 2006). We therefore introduce the auxiliary variables $\boldsymbol{m} = \{m_{jc} \mid \forall j \in \{1, \ldots, \bar{C}\}, \forall c \in \{1, \ldots, C\}\}$ that need to be sampled in order to compute $\boldsymbol{\beta}$. However, being $m_{jc}$ dependent on $n_{jc}$, the sampling step of these variables is very inefficient for large values of $n_{jc}$, as the probability values are proportional the Stirling number of the first-kind $s(n_{jc}, \cdot)$ (Fox et al., 2008). Luckily, we can avoid this step by observing that the value $m_{jc}$ corresponds to the number of tables where dish $q_u = c$ is served at restaurant $j$ in the Chines Restaurant Franchise (CRF) representation (Teh et al., 2006; Fox et al., 2007). Thus, we can compute each $m_{jc}$ by simply simulating the table assignments process. We recall that, in the CRF representation, each customer (i.e., observation) of each restaurant (i.e., group) is assigned to a table where just a single dish (i.e., mixture component) is served. Thus, while all customers sitting at the same table must be eating the same dish, there can be multiple tables serving the same dish as well.

Knowing that customer $u$ is eating the dish $q_u = c$, its table assignment $t_u$ can be sampled according to:

$$P(t_u = t \mid q_u = c, j_u = j, \boldsymbol{c}, \boldsymbol{t}^{-u}, \boldsymbol{\beta}, \alpha_0) \propto \begin{cases} \tilde{n}_{jt}^{-u}, & \forall t \text{ s.t. } c_{jt} = c; \\ \alpha_0 \beta_c, & t = t_{new}, \end{cases} \quad (5)$$

where $\boldsymbol{t}^{-u}$ represents the tables assigned to all the other nodes except $u$, $c_{jt} \in \boldsymbol{c}$ specifies the dish assigned to table $t$ at restaurant $j$ and $\tilde{n}_{jt}^{-u}$ denotes the number of customers (except $u$) sitting at table $t$ of restaurant $j$. Since we know the dish $q_u$ selected by the customer $u$, there is zero probability that the customer sits to a table where that dish is not served. The creation and deletion of tables is very similar to that of Eq. 4, so we skip it in the interest of the exposition and refer to the pseudocode in Appendix B for a complete treatment.

At last, after computing $m_{jc}$ as described above (i.e., $\sum_{t'} \mathbb{I}[c_{jt'} = c]$), the base distribution $\boldsymbol{\beta}$ is sampled from:

$$\boldsymbol{\beta} \mid \boldsymbol{q}, \boldsymbol{m} \sim \text{Dir}(\sum_{j=1}^{\bar{C}} m_{j1}, \ldots, \sum_{j=1}^{\bar{C}} m_{jC}, \gamma), \quad (6)$$

where Dir stands for the Dirichlet distribution.

**Sampling $\boldsymbol{\theta}$.** To update the emission parameters $\boldsymbol{\theta}$, we rely on its posterior given $\boldsymbol{q}$ and $\boldsymbol{x}$:

$$P(\boldsymbol{\theta}_c \mid \boldsymbol{q}, \boldsymbol{x}) \propto h(\boldsymbol{\theta}_c) \prod_{\forall u \mid q_u = c} f(x_u \mid \boldsymbol{\theta}_c). \quad (7)$$

By choosing the family of the base distribution $\boldsymbol{H}$ to be a conjugate prior for $F$, e.g., a Dirichlet distribution for Categorical emissions or a Normal-Gamma distribution for Normal emissions, we can compute the posterior in closed form.

Let the emission distribution be a categorical distribution with $M$ possible states. When creating a new state, we can sample the emission parameter according to a Dirichlet distribution, which is a conjugate prior for the categorical distribution:

$$\theta_c \sim \text{Dir}(\eta, \ldots, \eta), \tag{8}$$

where the subscript $c$ indicates the mixture component. Thanks to the conjugate prior, the emission parameters can be updated by sampling its Dirichlet posterior distribution:

$$\theta'_c \sim \text{Dir}(\eta + N_c^1, \ldots, \eta + N_c^M), \tag{9}$$

where $N_c^m$ indicates the number of times the visible label $m$ has been associated with the latent state $c$, i.e., $N_m^c = \sum_u \mathbb{I}[q_u = c \wedge x_u = m]$.

Similarly to the categorical case, let the emission distribution be an univariate Gaussian. In this case, for each state, we can sample the emission parameter according to a Normal-Gamma distribution:

$$\mu_c \sim \mathcal{N}(\mu_0, 1/(\lambda_0 \tau_c)) \tag{10}$$

$$\tau_c \sim \text{Gamma}(a_0, b_0), \tag{11}$$

where the subscript $c$ indicates a mixture component ant $\tau_c$ is the inverse of the variance. Then, the emission parameters of the Gaussian can be updated as follows:

$$\mu'_c \sim \mathcal{N}\left(\frac{\lambda_0 \mu_0 + N_c \bar{x}_c}{\lambda_0 + N_c}, \frac{1}{(\lambda_0 + N_c)\tau'_c}\right) \tag{12}$$

$$\tau'_c \sim \text{Gamma}\left(a_0 + \frac{N_c}{2}, b_0 + \frac{1}{2}\left(N_c s_c + \frac{\lambda_0 N_c (\bar{x}_c - \mu_0)^2}{\lambda_0 + N_c}\right)\right), \tag{13}$$

where $N_c$ indicates the number of visible labels associated with the latent state $c$ (i.e., $N_c = \sum_u \mathbb{I}[q_u = c]$), $\bar{x}_c$ is the mean of the data associated with the class $c$ (i.e., $\bar{x}_c = \frac{1}{N_c}\sum_{\forall u|q_u=c} x_u$), and $s_c$ is the variance of the data associated with the class $c$ (i.e., $s_c = \frac{1}{N_c}\sum_{\forall u|q_u=c}(x_u - \bar{x}_u)^2$).

**Sampling $\alpha_0$.** Following (Teh et al., 2006), the concentration parameter $\alpha_0$ can be updated between Gibbs sampling iterations by exploiting an auxiliary variable schema. Assume that $\alpha_0$ has a Gamma prior distribution $\text{Gamma}(a, b)$ (i.e., $\alpha_0 \sim \text{Gamma}(a, b)$). Then, we define the auxiliary variables $w_1, \ldots, w_{\bar{C}}$ and $s_1, \ldots, s_{\bar{C}}$, where each $w_j$ variable takes a value between 0 and 1, and each $s_j$ is a binary variable. Then, the value of $\alpha_0$ can be sampled according to the following schema:

$$w_j \sim \text{Beta}(\alpha_0 + 1, n_{j.}), \tag{14}$$

$$s_j \sim \text{Bernoulli}\left(\frac{n_{j.}}{n_{j.} + \alpha_0}\right), \tag{15}$$

$$\alpha_0 \sim \text{Gamma}\left(a + m_{..} - \sum_{j=1}^{\bar{C}} s_j, b - \sum_{j=1}^{\bar{C}} \log w_j\right), \tag{16}$$

where $n_{j.}$ is the number of costumer eating in the $j$-th restaurant, and $m_{..}$ is the total number of tables in all the restaurants.

**Sampling $\gamma$.** Similarly, assuming that the hyperparameter $\gamma$ has a gamma prior distribution $\text{Gamma}(a', b')$ (i.e., $\gamma \sim \text{Gamma}(a', b')$), its value can be updated by following the auxiliary variable schema below (Teh et al., 2006; Fox et al., 2008):

$$r \sim \text{Beta}(\gamma + 1, m_{..}), \tag{17}$$

$$p \sim \text{Bernoulli}\left(\frac{m_{..}}{m_{..} + \gamma}\right), \tag{18}$$

$$\gamma \sim \text{Gamma}(a' + C - p, b' - \log r). \tag{19}$$

## B  ICGMM PSEUDOCODE

To facilitate the practical understanding of our model, we provide the pseudocode of the Gibbs sampling method employed in this work.

---

**Algorithm 1** Gibbs sampling method for exact ICGMM

---

**Require:** A dataset of graphs $\mathcal{D} = \{g_1, \dots, g_N\}$. Initialize $C = 1$, $\boldsymbol{\theta} = \{\theta_1\}$ (where $\theta_1 \sim H$), $\mathcal{T}_j = \emptyset$ (for all restaurant $j$), $\boldsymbol{q} = \boldsymbol{t} = \boldsymbol{c} = \perp$, and $\boldsymbol{n} = \tilde{\boldsymbol{n}} = \boldsymbol{0}$.

> **repeat**
>> **for** $g \in \mathcal{D}$ **do**                                           ▷ For each graph
>>> **for** $u \in \mathcal{V}_g$ **do**                                        ▷ For each node
>>>> *// assign the restaurant*
>>>> $j_u \leftarrow \psi(\mathbf{q}_{\mathcal{N}_u}^{\ell-1})$                ▷ Can be done once $\forall u$
>>>>
>>>> *// assign the dish*
>>>> $n_{j_u q_u} \leftarrow n_{j_u q_u} - 1$                                   ▷ If $q_u \neq \perp$, remove $q_u$ from the counting
>>>> $q_u \leftarrow \text{SAMPLING}(j_u, \boldsymbol{n}, \boldsymbol{\theta}, \boldsymbol{x}, \boldsymbol{\beta}, \alpha_0)$   ▷ Sample the dish according to Eq. (4)
>>>> **if** $q_u$ is new **then**                                              ▷ Create a new state
>>>>> $\theta_{\text{new}} \sim H$
>>>>> $\boldsymbol{\theta} \leftarrow \boldsymbol{\theta} \cup \{\theta_{\text{new}}\}$
>>>>> $C \leftarrow C + 1$
>>>>> $n_{j q_u} \leftarrow 0 \quad \forall j \in \{1, \dots, \bar{C}\}$       ▷ Initialize the counters
>>>>
>>>> **end if**
>>>> $n_{j_u q_u} \leftarrow n_{j_u q_u} + 1$                                   ▷ Update the counter
>>>>
>>>> *// assign the table*
>>>> $\tilde{n}_{j_u t_u} \leftarrow \tilde{n}_{j_u t_u} - 1$                   ▷ If $t_u \neq \perp$, remove $t_u$ from the counting
>>>> $t_u \leftarrow \text{SAMPLING}(j_u, q_u, \boldsymbol{c}, \tilde{\boldsymbol{n}}, \boldsymbol{\beta}, \alpha_0)$   ▷ Sample the table according to Eq. (5)
>>>> **if** $t_u$ is new **then**                                             ▷ Create a new table
>>>>> $\mathcal{T}_j \leftarrow \mathcal{T}_j \cup \{t_u\}$
>>>>> $c_{j_u t_u} \leftarrow q_u$                                            ▷ Save the dish-table assignment
>>>>> $m_{j_u q_u} \leftarrow m_{j_u q_u} + 1$                                ▷ Update the table count
>>>>> $\tilde{n}_{j_u t_u} \leftarrow 0$                                      ▷ Initialize customer counter
>>>>
>>>> **end if**
>>>> $\tilde{n}_{j_u t_u} \leftarrow \tilde{n}_{j_u t_u} + 1$
>>> **end for**
>> **end for**
>>
>> *// remove unused dishes*
>> **for** $c \in \{1, \dots, C\}$ **do**
>>> **if** $\sum_{j=1}^{\bar{C}} n_{jc} = 0$ **then**                          ▷ No customers eat the dish $c$
>>>> $\boldsymbol{\theta} \leftarrow \boldsymbol{\theta} \setminus \{\theta_c\}$
>>>> $C \leftarrow C - 1$
>>> **end if**
>> **end for**
>>
>> *// remove empty tables*
>> **for** $j \in \{1, \dots, \bar{C}\}$ **do**
>>> **for** $t \in \mathcal{T}_j$ **do**
>>>> **if** $\tilde{n}_{jt} = 0$ **then**                                      ▷ No customers eat at the table $t$ in the restaurant $j$
>>>>> $\mathcal{T}_j \leftarrow \mathcal{T}_j \setminus \{t\}$
>>>>> $m_{j c_{jt}} \leftarrow m_{j c_{jt}} - 1$
>>>> **end if**
>>> **end for**
>> **end for**
>>
>> *// update model parameters*
>> $\boldsymbol{\beta} \leftarrow \text{SAMPLING}(\boldsymbol{q}, \boldsymbol{m})$   ▷ Sample according to Eq. (6)
>> $\boldsymbol{\theta} \leftarrow \text{SAMPLING}(\boldsymbol{q}, \boldsymbol{x})$   ▷ Sample according to Eq. (7)
>>
>> **if** ICGMM$_{auto}$ **then**
>>> $\alpha_0 \leftarrow \text{SAMPLING}(a, b, \boldsymbol{n})$                ▷ Sample according to Eq. (14), (15), (16)
>>> $\gamma \leftarrow \text{SAMPLING}(a', b', \boldsymbol{m})$               ▷ Sample according to Eq. (17), (18), (19)
>> **end if**
> **until** stopping criteria

---

## C  DATASET STATISTICS

Below we report some statistics for the chosen benchmarks.

Table 3: Dataset statistics.

|  |  | # Graphs | # Classes | # Nodes | # Edges | # Node labels |
|---|---|---|---|---|---|---|
| CHEM. | DD | 1178 | 2 | 284.32 | 715.66 | 89 |
| | NCI1 | 4110 | 2 | 29.87 | 32.30 | 37 |
| | PROTEINS | 1113 | 2 | 39.06 | 72.82 | 3 |
| SOCIAL | IMDB-BINARY | 1000 | 2 | 19.77 | 96.53 | - |
| | IMDB-MULTI | 1500 | 3 | 13.00 | 65.94 | - |
| | REDDIT-BINARY | 2000 | 2 | 429.63 | 497.75 | - |
| | REDDIT-5K | 4999 | 5 | 508.82 | 594.87 | - |
| | COLLAB | 5000 | 3 | 74.49 | 2457.78 | - |

## D  SPEEDUP GAINS WITH FASTER INFERENCE

We compare the performances of the exact version of ICGMM against the faster implementation. As we can see, the speedup increases for the datasets with larger average number of nodes (see Table 3).

Table 4: Approximate speedup between the exact ICGMM and the faster version on all datasets.

|  |  | ICGMM | ICGMM$_f$ |
|---|---|---|---|
|  |  | ref. | min/max |
| CHEM. | DD | 1× | 17.8×/30.8× |
| | NCI1 | 1× | 3.1×/5.1× |
| | PROTEINS | 1× | 4.2×/5.7× |
| SOCIAL | IMDB-B | 1× | 2.4×/5.1× |
| | IMDB-M | 1× | 1.6×/3.6× |
| | REDDIT-B | 1× | 11.1×/45.6× |
| | REDDIT-5K | 1× | 36.7×/60.6× |
| | COLLAB | 1× | 3.1×/8.6× |

