# OpenReview forum: "The Infinite Contextual Graph Markov Model"
_ICLR.cc/2022/Conference — ICLR 2022 Submitted_

### Official Review · Reviewer_P2Q1 · 2021-11-02

**Correctness:** 3
**Technical Novelty And Significance:** 3
**Empirical Novelty And Significance:** 2
**Recommendation:** 5
**Confidence:** 2

**Main Review:**

The authors propose an interesting idea of utilizing the HDP framework to obtain the number of latent variables at each layer. HOwever the gain in the accuracy seems either non-existent or marginal.

**Summary Of The Paper:**

The paper combines the idea from the classical hidden dirichlet processe and CGMM to automate the process of selecting the hyperparameters of CGMM.

**Summary Of The Review:**

Interesting idea but not very impressive experimental results.

---

### Official Review · Reviewer_aqe6 · 2021-11-02

**Correctness:** 3
**Technical Novelty And Significance:** 3
**Empirical Novelty And Significance:** 2
**Recommendation:** 5
**Confidence:** 3

**Main Review:**

Strengths:
1. The paper is well written and easy to follow.
2. Selecting an optimal model size (trainable parameters of each neural network) is essential to obtain reasonable performance. However, it involves an expensive model selection procedure. This work investigates automatic model size selection without compromising model performance.
3.  Experimenting with chemical and social network datasets, the authors show the method perform comparable performance with the state-of-the-art baselines for graph classification problem.


Weakness:
1. However, the proposed method is restricted to a specific type of architecture, which is the Contextual Graph Markov model. Though the authors discussed why they have selected this architecture, it somewhat limits the applicability of the approach.  Other than CGMM, there exist powerful adaptive graph representation techniques like GraphSage. Can their proposed method be extended to automate model size selection for those neural models?

2. Please discuss how likely it is to be extended to unsupervised generative model-based (VAE) techniques to obtain a representation of a graph?

3. The authors should provide some comparison with AutoGNN /DGN types of methods in terms of performance accuracy and time of execution.

4. The authors should provide execution time comparison with all existing baselines instead of only their model. Because the time overhead should not be significantly high to reduce the model selection overhead.

5. They should provide the analysis of graph size versus the size of the latent variable, is there any interesting pattern or observation?

**Summary Of The Paper:**

In this paper, the authors propose a mechanism to automate the size selection of each latent representation layer of the Contextual Graph Markov model. The model automatically adjusts the size of the model parameters mitigating the expensive model selection. Moreover, the authors introduce some techniques to scale the
proposed solution.

**Summary Of The Review:**

This work investigates automatic model size selection without compromising model performance, which is a very important and timely research question.

However, the proposed solution revolves around a specific architecture that limits its applicability. Also, it does not consider the edge properties of graphs, which are extremely important for different graphs like crystal (CGCNN - Phys. Rev. Lett.).

More model ablation studies for e.g graph size (or structural complexity like wiener index/diameter/density) vs learned latent variable size are required.

---

### Official Review · Reviewer_ynws · 2021-11-02

**Correctness:** 4
**Technical Novelty And Significance:** 4
**Empirical Novelty And Significance:** 4
**Recommendation:** 8
**Confidence:** 3

**Main Review:**

The proposed method is very interesting, since it's an unsupervised method for automatic selection of network structure, as opposed to the more popular supervised AutoML techniques. Traditional inference mechanisms for BNP-based methods are costly to run on larger datasets, hence authors propose a faster Gibbs sampling-based inference method.

The proposed method performs impressively against supervised learning-based methods and the parametric version of the method, i.e. CGMM. Given the unsupervised nature of the method, the results are encouraging for future work in the direction of BNPs.

I lean towards acceptance of this paper, as it has the potential to encourage research on the integration of BNP methods with Deep Neural Networks.

**Summary Of The Paper:**

Authors propose a Bayesian Non-Parametric (BNP) method to automatically learn the "structure" of the graph neural network. Specifically, the authors have proposed a Hierarchical Dirichlet Process model for the automatic inference of latent code's size.

**Summary Of The Review:**

The authors propose an unsupervised Bayesian Non-Parametric method for GNN, which learns the GNN structure in an unsupervised fashion. For scaling-up, the authors propose a faster Gibbs Sampling based inference method. Experimental results indicate the strong performance of the method as compared to supervised learning methods. Overall, the direction is very interesting and could inspire future work at the intersection of BNP and Neural Networks.

---

### Official Review · Reviewer_mmkq · 2021-11-04

**Correctness:** 4
**Technical Novelty And Significance:** 2
**Empirical Novelty And Significance:** 2
**Recommendation:** 5
**Confidence:** 4

**Main Review:**

Strengths:

1. The idea of using Hierarchical Dirichlet Processes is quite intuitive.

The key idea of the paper is to use HDP to automatically decide the size of latent representations in the Contextual Graph Markov Model. To me, the idea of using HDP is intuitive.

2. The proposed model is theoretically sound.

For Hierarchical Dirichlet Processes, one critical problem is inference, and the authors use Gibbs sampling to deal with that as described in appendix, and I think the derivation is theoretically sound.

Weaknesses:

1. The novelty of the paper is insufficient.

The key idea is on using HDP in the Contextual Graph Markov Model. However, the idea is not new, as it is a quite common practice to use HDP in unsupervised generative models for automatic model selection. For example, in clustering, HDP has been widely used to automatically decide the number of clusters. Similarly in topic modeling, many works use HDP to decide the number of latent topics. Given these well-known efforts in machine learning, I think the idea of extending the Contextual Graph Markov Model with HDP is not so innovative.

Also, this paper uses Gibbs sampling for inference, which has also been widely used for HDP, and the Gibbs sampling procedure derived in this paper is not so different from those used in existing works. In this sense, this paper doesn't provide many new insights or ideas in terms of model inference.

Therefore, although the paper has some interesting ideas, I think the overall novelty is insufficient.

2. The improvement over existing methods is not significant.

In the experiment, the authors compare iCGMM against many existing methods on the task of graph classification. However, on chemical datasets, the results of iCGMM are worse than a few baseline methods (e.g., GIN and DiffPool). On social datasets, the improvement over these baseline methods is also insignificant, given the high standard deviation. In particular, iCGMM only slightly outperforms CGNN, making the advantage of applying HDP to CGMM less convincing.


**Summary Of The Paper:**

This paper is an extension of the Contextual Graph Markov Model, a deep unsupervised probabilistic approach for modeling graph data. The key idea is to leverage Hierarchical Dirichlet Processes, which enables the proposed approach to automatically choose the size of each layer’s latent representation. The authors conduct experiment on graph classification tasks, and the results are quite promising.


**Summary Of The Review:**

In summary, this paper applies HDP to CGMM, which is intuitive and theoretically sound. However, the idea of using HDP is not new in unsupervised generative models, and the improvement of iCGMM over CGMM seems insignificant. Thus, I would lean towards a weak reject.

---

### Decision · Program_Chairs · 2022-01-20

**Decision:**

Reject

**Comment:**

This paper extends the Contextual Graph Markov Model, a deep unsupervised probabilistic approach. The key idea is to leverage Hierarchical Dirichlet Processes to automatically determine each layer's latent representation's size. The paper conducts experiments on graph classification tasks to show the superiority of the proposed method.

Strength
* A new method is proposed.
* The proposed method appears to be sound.
* Experiments are conducted to demonstrate the effectiveness.

Weakness
* The novelty and significance of the work are not enough.
* The improvements on existing methods are not significant.
* The proposed method is also not so general.

-----------

After rebuttal

Reviewer ynws, who gave the highest score, says

“I agree with the overall review of the paper by other reviewers. The proposed method is limited to the CGMM model and not generic enough to extend to other more popular graph neural networks. The improvements don't seem to be significant enough as well.”